# Detection of Biomarkers Relating to Quality and Differentiation of Some Commercially Significant Whole Fish Using Spatially Off-Set Raman Spectroscopy

**DOI:** 10.3390/molecules25173776

**Published:** 2020-08-19

**Authors:** Jeremy D. Landry, Peter J. Torley, Ewan W. Blanch

**Affiliations:** 1Applied Chemistry and Environmental Science, School of Science, RMIT University, 124a La Trobe Street, Melbourne, VIC 3000, Australia; 2Biosciences and Food Technology, School of Science, Building 223, RMIT University, P.O. Box 71, Bundoora, VIC 3083, Australia; peter.torley@rmit.edu.au

**Keywords:** Raman spectroscopy, spatially offset Raman, fish muscle, chemometrics

## Abstract

Aquaculture represents a major part of the world’s food supply. This area of food production is developing rapidly, and as such the tools and analytical techniques used to monitor and assess the quality of fish need to also develop and improve. The use of spatially off-set Raman spectroscopy (SORS) is particularly well-suited for these applications, given the ability of this technique to take subsurface measurements as well as being rapid, non-destructive and label-free compared to classical chemical analysis techniques. To explore this technique for analysing fish, SORS measurements were taken on commercially significant whole fish through the skin in different locations. The resulting spectra were of high quality with subsurface components such as lipids, carotenoids, proteins and guanine from iridophore cells clearly visible in the spectra. These spectral features were characterised and major bands identified. Chemometric analysis additionally showed that clear differences are present in spectra not only from different sections of a fish but also between different species. These results highlight the potential application for SORS analysis for rapid quality assessment and species identification in the aquaculture industry by taking through-skin measurements.

## 1. Introduction

Fish are a major food source for much of the world. The growth of the global aquaculture industry has been steady for a number of years as capture fishing begins a slow decline [1]. As with all areas in food production, chemical and physical parameters need to be measured to ensure that the product meets the expectations of consumers. Fat content, flesh colour and thaw cycle are important measures used to determine overall fish quality.

Typically, these types of analysis are performed using classical methods such as gas chromatography [2], high pressure liquid chromatography [3] and colourmetric [4] or colour card systems [5]. While these analyses are robust, accurate and widely accepted as the “gold standard” practice in analytical science, for commercial purposes they can be difficult to employ due to their expensive set-up and running costs and lengthy analysis times which can bottleneck production. 

Raman spectroscopy offers the ability to perform rapid, non-destructive, label-free chemical analysis, which is a huge advantage in food analysis. This form of vibrational spectroscopy depends upon observing the effects of inelastic Raman scattering, with energy being transferred from the photon to the molecule (known as Stokes scattering), or from the molecule to the photon (known as anti-Stokes scattering) [6]. Raman spectroscopy measures the intensity of the scattered radiation, with the frequencies of scattered photons generating Raman spectral bands. Raman bands that have a lower frequency than the incident radiation are referred to as Stokes bands and those that have a higher frequency are called anti-Stokes bands. Stokes Raman scattering is more intense than anti-Stokes Raman scattering, and so it is more commonly used. The frequency of each Raman band in the spectrum corresponds to a vibrational mode of the molecule, and the scattered photon frequency is sensitive to the constituent atoms and bonds, or chemical functional groups, involved in the corresponding vibrational mode, as well as being influenced by their local environment. Raman spectroscopy is sensitive to chemical composition and has inherent advantages for rapid and non-invasive characterisation of complex samples, such as foods.

Several biomarkers relating to fish quality have been previously detected in fish using Raman spectroscopy and related vibrational techniques such as Infrared Spectroscopy. These include biomarkers for lipids [7,8], carotenoids [9,10], the effects of freeze–thaw cycles [11] and species identification and authentication [12]. While these examples of biomarker detection were made much faster in comparison to their detection using classical techniques and produced comparable results, standard Raman spectroscopy measurements on fish samples still require the fish to be processed in some way, most commonly by filleting. This is time consuming and is obviously highly invasive and fatal to the animal. Thus, conventional Raman spectroscopic studies of fish tissues beneath the skin are limited in their application. 

A potential solution is presented by a relatively recent technological development of Raman spectroscopy that makes it possible to measure the chemical composition of biological tissues at sub-surface levels. This innovative Raman technique is known as spatially off-set Raman spectroscopy (SORS), and it enables the time-dependent selection of Raman-scattered photons at different distances beneath the surface of a sample. As such, SORS has proven to be able to provide reliable biochemical information on both the surface and sub-surface levels of complex samples and has particular potential in this space due to its ability to take measurements through surface layers. SORS detects Raman scattering that occurs away from the position of the incident laser spot on the sample surface and mathematically isolates the subsurface spectra [13]. Initial studies of SORS in food analysis have proven to be successful in identifying important biomarkers in muscle foods [14], and have been used to assess the viability of taking measurements through fish skin, although this earlier study was not performed on whole fish [15]. 

The aim of this project was to determine the viability of SORS to detect biological markers relating to fish quality through the skin in whole fish. A particular objective was to test the ability of SORS to detect lipid and pigmentation biomarkers, as these are of significant importance in aquaculture as measures of fish quality and health. A small survey group of commercial fish was chosen in order to test the sensitivity of SORS spectra to these biomarkers, however this is still sufficient for testing the feasibility of SORS as a biomolecular characterization technique for aquaculture and fisheries.

## 2. Materials and Methods 

### 2.1. Samples

The following fish samples were selected: Atlantic salmon (*Salmo salar L.*) was provided by Petuna Seafoods, while rainbow trout (*Oncorhynchus mykiss*), Asian seabass (*Lates calcarifer*) and red seabream (*Pagrus major*) were purchased at the Queen Victoria Markets in Melbourne. All fish were acquired as fresh, head on, gutted and without being descaled. Prior to measurements, the whole fish were wrapped in cling wrap. SORS spectral analysis was conducted in September 2019 on the same day as sample acquisition to minimise the effects of ageing and oxidation. 

These four species of fish were selected for the following reasons. Trout and salmon are both salmonids, and thus are both fatty fish, are both widely farmed around the world in aquaculture and have flesh pigmentation affected by carotenoids in the diet. Asian seabass is a widely farmed fish in Australia but is considered a lean fish without pigmentation. Red seabream was chosen for its skin colour as it is more brightly coloured than many other commercially significant fish types, and additionally is also considered a lean fish. Therefore, the Asian seabass and red seabream samples can be considered as extreme limits of carotenoid pigmentation in common commercial fish species. Together, these four fish samples provide a small but relevant sample set for testing the sensitivity of SORS spectra to the biochemistries of commercial fish species. Furthermore, we have specifically studied commercially sourced fish samples as these are directly representative of standard fish that are available to, and consumed by, the general public. Any conclusions drawn from these measurements are, thus, expected to be generally applicable to these fish species when sourced from commercial aquaculture industries, and are representative of the biochemistry of fish consumed by the general public. 

### 2.2. Raman Measurements 

Spatially-offset Raman spectroscopy measurements were made using an Agilent RapID Raman System. Spectral acquisitions were taken using 0.4 s maximum exposure time and 0.8 s offset maximum exposure time. The laser excitation wavelength was 830 nm at 1 cm^−1^ spectral resolution with a laser power strength of 450 mW at the source (5 mW at the sample), which was used for all measurements. The SORS algorithm was set to Baselined Offset, with the baseline type set to none.

The instrument requires a pressure interlock located on the front of the probe to be engaged before a measurement can be taken. The probe was held at an approximate 90° angle to the sample, with the probe making direct contact with the sample. Once in position, the probe only needed to be steadied by hand to remain in position, while the weight of the probe engaged the interlock and allowed a measurement to be taken. This process maintained a consistent pressure from the probe against the sample for all measurements made. 

Scans were taken in four different locations on the right hand side of each fish that are recognised as being important for assessing fish quality; these locations, as shown in Figure 1, are the arterial dorsal region, belly flap, the mid-section also known as the Norwegian quality cut (NQC) in the salmon industry and the tail region. Twenty five (25) replicate scans were taken at each of the four regions for each fish. Each of the fish species were analysed in duplicate. Sampling regions were covered in clingfilm before taking measurements. The RapID probe was moved by hand to take scans at different locations within each region of the fish.

### 2.3. Spectra Pre-Processing and Chemometrics

All spectra were pre-processed using the PLS toolbox package in MatLab. The 25 spectra measured at each site were averaged. Smoothing was performed using the Savitsky–Golay filtering (order 2, framelen 3) and baseline corrected (order 4). Spectra normalization was performed using the average peak height of the entire spectra for the range from 200 to 1800 cm^−1^.

Principal component analysis was performed using The Unscrambler 11.0 package with a standard value decomposition algorithm. Spectral regions below 1000 cm^−1^ and above 1750 cm^−1^ were excluded from the principal component analysis to reduce distortion effects from Rayleigh scattering and baselining. 

## 3. Results

### 3.1. Spectra and Band Assignments

Figure 2 shows the mean processed spectra from each of the four fish species analysed and overlayed on top of each other. Significant differences are obvious between these averaged spectra for the four examined species. Guanine peaks at 650 and 930 cm^−1^ were observed in the spectra of all species. In Figure 2, strong bands associated with lipids [7] can be observed at 968, 1065, 1078, 1266, 1300, 1440 and 1750 cm^−1^. The Amide I band profile is clearly visible with a peak maximum at 1660 cm^−1^, and there are distinct bands from collagen at 724 and 868 cm^−1^. Bands at 1550 and 1608 cm^−1^ are typically assigned to two prominent amino acids, tryptophan and phenylalanine [16], while Raman peaks associated with carotenoids are also visible at 1004, 1160, 1192 and 1520 cm^−1^. All other bands visible in the spectra are described in further detail in Table 1. Spectra from the individual regions can be found in the Appendix A. 

### 3.2. Chemometrics

Principal component analysis was performed on these measured spectra in order to enhance discrimination between their features with respect to species origin, with an example of this analysis being shown in Figure 3 in the form of a three dimensional plot with the three most significant principal components plotted orthogonally. Individual SORS measurements are represented by the individual data points and are colour-coded as shown in the figure legend. Clear clustering of the SORS spectra principal components based upon fish species is evident, verifying that these three principal components are sensitive to biochemical differences between these four species. While rotation of this figure around any axis can emphasise different perspectives, all other orientations confirmed the obvious clustering of these spectra. This PCA clustering shows that SORS spectra measured through the skin are able to identify biochemical signatures within the tissues that can be used to differentiate between the four fish species investigated. This result also leads to the next step of investigating which biochemical signatures in the SORS spectra are important for differentiating between these fish species.

The three most significant loadings for the principal component analysis are presented in Figure 4. Principal component 1 shows strong negative bands at 1330, 1440 and 1660 cm^−1^ which all relate to lipids. The second principal component displays several strong positive bands that relate to guanine, these being found at 1235, 1360, 1390 and 1420 cm^−1^. In addition, there are two negative bands at 1160 and 1520 cm^−1^ that arise from carotenoids present in fish muscle and skin. We note the opposing signs of the guanine and carotenoid bands and that this appears to reflect a contra-correlation of the occurrence of these two biomolecules in different tissues. Several features well above background noise are also observed for principal component 3, including the positive feature at 1235 cm^−1^ from guanine. 

## 4. Discussion

### 4.1. Lipids

Lipid bands can be seen in all of the four sampled regions for each fish species examined. Dorsal regions show the lowest intensity while the lipid bands are much more intense for the belly region. This observation is in good agreement with fat distribution as seen using visual and infrared spectroscopy, where fat distribution in fish fillets can be seen to be high in the belly where much of the flesh is adipose tissue, while the dorsal region is much more lean [19]. 

Variation in the lipid SORS band intensity can also occur as a consequence of the distribution of fat within each region of the fish. Atlantic salmon, rainbow trout and red seabream flesh are striated with tissue regions known as myosepta that separate each of the individual muscle fibres. These regions are found in both the dark and light muscle regions, which are respectively located near the skin surface and closer towards the internal structure of the fish. The myosepta are known to contain large amounts of lipids, with 39% of lipids found within the myosepta in the white muscle region and 62% in the dark muscle regions [20,21]. Considering that some myosepta run close to the skin surface, it is possible that some SORS measurements may focus directly on these points, essentially “spiking” the spectra with higher intensity lipid peaks. 

Of the four fish species studied here, differences in lipid content and fatty acid profiles are well known. All trout and salmon species are regarded as high-fat fish, with total lipid content coming in at above 10%. Conversely, red seabream and Asian seabass are considered as very low-fat fish with less than 2% by weight lipid content [22]. As can be seen in the loadings in Figure 4, lipid-associated bands are a major source of difference between the SORS spectra for each of the fish species, as all the major lipid-associated bands are present in principal component 1. This effect is most obvious in the PCA plots shown in Figure 3, with both of the high-fat-content fish scoring positively for principal component 1. 

As a comment on the potential usefulness of SORS spectra for sensing lipids, the major peaks between 1000 and 1800 cm^−1^ have been used to determine levels of unsaturation of fats in fish tissue [8] and quantify total and individual omega-3 fatty acids in commercial fish oils [23]. While differentiation and quantification of fatty acids from the lipoproteins in whole fish is complex and challenging, the SORS spectra obtained provide similar levels of intensities as those obtained in these studies. As these fatty acids are often used as the basis of beneficial health claims for high-fat fish such as salmon and trout, quantifying these fatty acids as a quality measure of fish production would be beneficial to the aquaculture industry [24]. 

### 4.2. Guanine

A number of guanine bands are present in all four regions for all the fish presented in this study, particularly the bands at 650 and 930 cm^−1^. It is most likely that these guanine bands are originating from the iridophore cells, which are found in the sub-epidermal layer of fish, of which guanine is a major component [25]. Iridophore cells contain stacked plates of crystalline chemochromes consisting primarily of guanine that refract light, leading to pigmentation in fish and many other animals.

There are relative differences in the guanine band intensities between the four fish species investigated here, however it is not yet clear if this due to the size, shape, thickness of the scales or due to the concentration of guanine in the iridophore cells present below the skin surface. The belly region for each of the fish has the most numerous and intense guanine bands. The NQC and tail regions both show relatively fewer guanine bands though with similar intensities, with the dorsal region having significantly less intensity and fewer discernible guanine-related bands. This finding appears to be consistent across all fish samples included in this study. Analysis of fish skin performed in previous studies shows that the concentration of guanine is lower in the dorsal region compared with the concentration found in the ventral region [26]. This difference in concentration could explain the guanine band intensity differences between the dorsal and both the NQC and tail regions, and also why the NQC and tail regions have similar intensities as both regions are located within the ventral section of fish. Considering that the belly region has the most guanine bands and intensity, it may be reasonable to suggest that guanine concentration further increases towards the belly in fish skin.

### 4.3. Carotenoids

Clear Raman bands associated with carotenoids appear in the spectra of each fish. Trout and salmon samples are well known to contain high levels of the carotenoid astaxanthin, as this carotenoid is responsible for the characteristic pigment found in their flesh [27]. The four bands observed at 1004, 1160, 1192 and 1520 cm^−1^ are in good agreement with bands reported for astaxanthin in the literature [28]. According to the spectral references for this compound, the two major bands in terms of intensity appear at 1160 and 1520 cm^−1^.

These two bands are very clear in the SORS spectra for rainbow trout and Atlantic salmon samples studied here, though these bands do not appear to be present in the spectra measured on the belly region for the salmon. However, when we consider that this region is composed of adipose tissue and that astaxanthin in Atlantic salmon is bound to muscle proteins, this may explain the lack of these bands in the SORS spectra from this region of the salmon [29]. The SORS bands ascribed to astaxanthin also appear weakly in the spectra for red seabream as this species is known for its red skin that is a result of astaxanthin inclusion. Furthermore, the intensity of these bands is strongest in the dorsal region where the skin pigmentation level is highest.

### 4.4. Proteins and Amino Acids

Bands for the amino acid tryptophan appear in the spectra for each fish species included in this study. This ability of SORS to monitor tryptophan levels within the tissues of these fish is particularly noteworthy as this amino acid does not appear in high concentration for the red seabream, rainbow trout and Atlantic salmon samples compared to other amino acids [30,31,32]. This suggests that much of the intensity of the measured amide group mode vibrations from proteins originates from the collagen known to occur at high concentrations in fish skin and muscles, which is typically type I collagen [33]. As a number of different indicators of texture are related to collagen content, there is potential for SORS data to be used to predict eating quality of meats of all types, including fish [34]. 

## 5. Conclusions

The spectra reported in this work show a higher level of detail than previous Raman spectroscopic analyses on a variety of fish. Most significantly, these results demonstrate that SORS spectra can reveal important biochemical details about the tissues of fish with measurements being made through the skin, a significant advantage for potential future application of Raman spectroscopy in aquaculture. Our SORS measurements have yielded descriptive and semi-quantitative information on lipid, protein and carotenoid composition in fish tissues as well as other sub-surface features. Though the range of fish species investigated in this study was small, these results do show that SORS can non-invasively characterize the differences in biochemistries not only between different commercial fish species but also between commercially significant body regions within the same individual fish. The differences in these spectral features show that identification and authentication of fish samples using SORS has strong potential in the future.

## Figures and Tables

**Figure 1 molecules-25-03776-f001:**
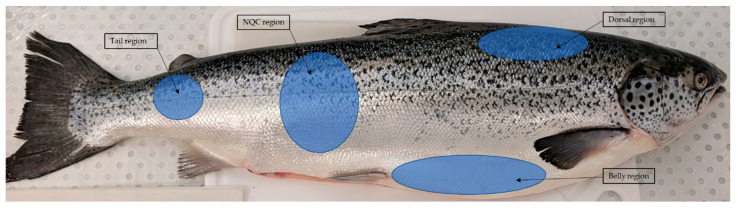
Location of each region where spectral measurements were taken.

**Figure 2 molecules-25-03776-f002:**
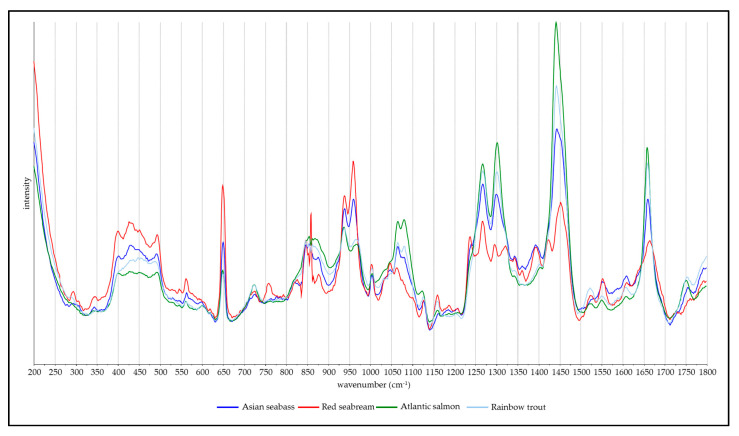
Mean processed spectra of each fish species.

**Figure 3 molecules-25-03776-f003:**
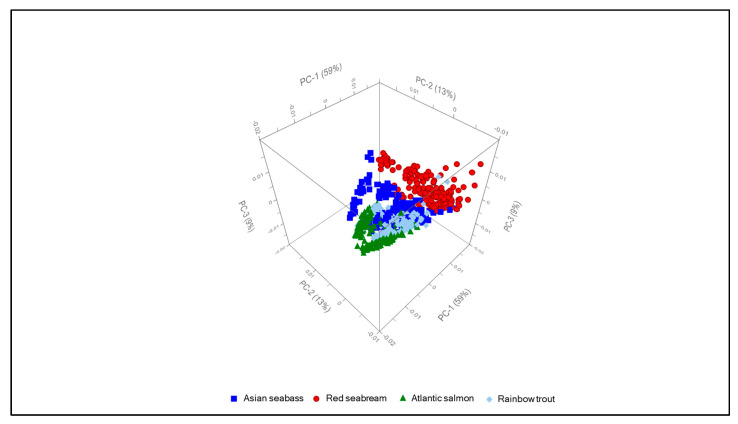
Principal component analysis of SORS (spatially off-set Raman spectroscopy) spectra. PC1 contributes 59% of the explained variance, with PC2 and 3 explaining 13% and 9%, respectively.

**Figure 4 molecules-25-03776-f004:**
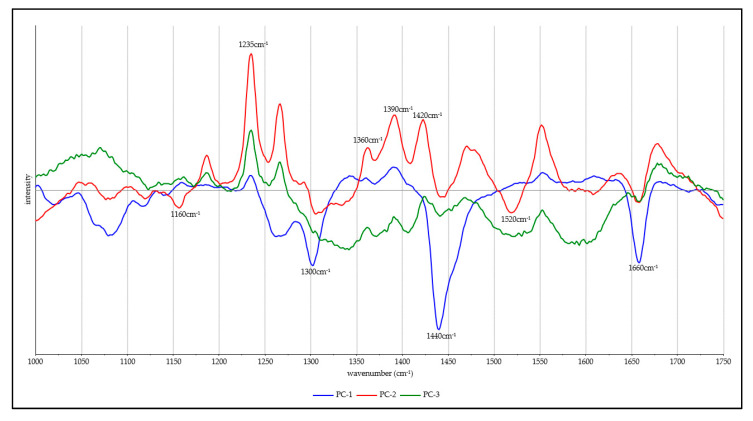
The three most significant spectra loadings from principal component analysis.

**Table 1 molecules-25-03776-t001:** Major band assignments of fish spectral features.

Wavenumber (cm^−1^)	Vibration	Component	Reference
339	δ(C=O)	Guanine	[17,18]
400	δ(C=O)	Guanine	
493	δ(N-9-C-4=C-5) and δ(N-7-C-=C-4)	Guanine	
560	δ(C-C=C)	Guanine	
650	Ring breathing	Guanine	
710	δ(ring)	Guanine	
724	C-C stretching	Collagen	[16]
760	Ring breathing	Tryptophan	
825	-	Collagen	
850	Single bond stretching	Amino acids	
860	ν(C-C)	Collagen	
868	ν(C-C) stretching	Collagen	
930	δ(N-C=N)	Guanine	[17,18]
968	δ(=C-H) trans RHC=CHR	Lipid	[7,8]
1004	In plane CH_3_ rock Aromatic ring breathing	Carotenoids Phenylalanine	[16]
1048	ν(C-H)	Lipids	[7,8]
1065	ν(C-C) stretching	Lipids	
1079	Asymmetric CCC stretching	Lipids	
1118	C-C stretching	Lipids	
1126	δ(C-N=C)	Guanine	[17,18]
1160	C-C polyene chain stretch	Carotenoids	[8,9]
1192		Carotenoids	
1235	ν(C-2-NH_2_)	Guanine	[17,18]
1266	C-H rocking	Lipid	
1300	CH_2_ twist	Methylene-Lipid	
1360	δ(C-8-N-H), δ(C-8-H), ν(C-8-N)	Guanine	[17,18]
1393		Guanine	
1420	δ(N-7=C-8-H)	Guanine	
1440	CH_2_ scissoring	Methylene-Lipid	[7,8]
1520	In phase C=C stretch	Carotenoids	[8]
1550	ν(C=C)	Tryptophan	[16]
1608	C=C	Phenylalanine	
1660	*cis* C=C stretch/amide I	Lipid/Protein	[7,8]
1750	C=O	Lipid	

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
