# Peer review of "Detection of Biomarkers Relating to Quality and Differentiation of Some Commercially Significant Whole Fish Using Spatially Off-Set Raman Spectroscopy"

_molecules, 2020, doi:10.3390/molecules25173776_

Round 1
Reviewer 1 Report
Authors proposed to use a commercial SORS to detect the quality of fish. It is quite interesting but there are several concerns which need to be clarified.
- The title is not fit with the content very well because actually the measurements were only taken at some points on the whole fish. So being titled as 'Characterisation of whole Fish' is not suitable.
- Again, the results are showing the differences between different species of fish. Therefore it is not characterizing the single fish itself.
- The background in the introduction is quite weak, especially for the SORS. Even though authors are using the commercial system, authors need to provide more introduction for the different kinds of SORS, for example, https://doi.org/10.1366/000370206778999102 or https://doi.org/10.1002/jbio.201700129
- The information related to the SORS system are also important, such as wavelength, laser power at the sample.
- During the measurements, the probe is hold by hand. My question is how to maintain the distance between the probe and sample? Even if they are touched with each other, the pressure will be different. These factors may affect the spectra.
- It was stated that analysis were performed between 1000 and 1750 cm^-1. The spectral region < 1000 cm^-1 is still far away from Rayleigh scattering. Other part of the draft is using those spectral regions.
Author Response
Reviewer 1
Authors proposed to use a commercial SORS to detect the quality of fish. It is quite interesting but there are several concerns which need to be clarified.
- The title is not fit with the content very well because actually the measurements were only taken at some points on the whole fish. So being titled as 'Characterisation of whole Fish' is not suitable.
Again, the results are showing the differences between different species of fish. Therefore it is not characterizing the single fish itself.
To reflect the above comments, the title of the paper has been changed to "Detection of biomarkers relating to quality and differentiation of some commercially significant whole fish using Spatially Off-set Raman Spectroscopy".
See lines 2 – 5.
- The background in the introduction is quite weak, especially for the SORS. Even though authors are using the commercial system, authors need to provide more introduction for the different kinds of SORS, for example, https://doi.org/10.1366/000370206778999102 or https://doi.org/10.1002/jbio.201700129
The introduction section has been expanded and now contains more background on Raman Spectroscopy and SORS along with more explanation on the application in detection of biomarkers in tissues and muscle foods.
See lines 36 – 75.
- The information related to the SORS system are also important, such as wavelength, laser power at the sample.
The laser excitation wavelength of 830 nm at typically 3 cm-1 spectral resolution and with a laser power of 450 mW”.
See lines 108-110.
- During the measurements, the probe is hold by hand. My question is how to maintain the distance between the probe and sample? Even if they are touched with each other, the pressure will be different. These factors may affect the spectra.
The instrument requires a pressure interlock located on the front of the probe to be engaged before a measurement can be taken. The probe was held at an approximate 90° angle to the sample, with the probe making direct contact with the sample. Once in position, the probe only needs to be steadied by hand to remain in position, while the weight of the probe engages the interlock and allow a measurement to be taken. This process maintained a consistent pressure from the probe against the sample.
See lines 111-116.
- It was stated that analysis were performed between 1000 and 1750 cm^-1. The spectral region < 1000 cm^-1 is still far away from Rayleigh scattering. Other part of the draft is using those spectral regions.
The PCA analysis only used the spectral region between 1000 and 1750 cm-1. The rest of the manuscript describes the features of the entire spectra.
Reviewer 2 Report
General remarks:
This article is well written and has interesting results, but in my opinion has a big flaw: exploratory works that use instrumental measurements for the first time in untested species, have to validate their results using classical chemical analysis techniques. Bibliographic information determined for one species do not directly apply to others, without analytical confirmation.
“Fish” are a very diverse group of organisms, with a lot of anatomical and physiological differences. Using conclusions from studies done on salmon to infer correlations on other species can be a bit implausible, especially if we are comparing species with completely different habitats, feeding habits or anatomical features. Fish cultivated in aquaculture continue to be morphologically, anatomically, and physiologically like their wild counterparts.
In my opinion these results are publishable, but the focus, depth and scope of the article must be changed.
To facilitate the location of my comments, I am going to present them organized by line number of the submitted pdf file.
Revision comments:
Line 2: The presented title does not describe adequately the work done in this study. When using the word “characterisation” is necessary to indicate what is going to be characterised. In this case the authors need to indicate what kind of molecules they are going to characterise. It is also necessary to indicate which species are going to be analysed in this work. The term “fish” includes a huge diversity of species, from bony fish to cartilaginous fish, and in this work only four bony species were analysed. The title should include the names of the species.
Line 28: The Introduction is very short and lacks topics necessary to fully understand the work! The fundamentals of Raman spectroscopy need to be described, along with the explanation on how it can be applied to determine specific molecules’ contents in skin and muscle tissues. The Introduction also lacks the advancements made by other authors in this field.
Line 30: The citation [1] is from 2018. FAO publishes annually the State of World Fisheries and Aquaculture. Authors should use the reference from 2020.
Line 33: The expression “to determine” is used twice.
Lines 42-43: The aim of this study needs to explicitly indicate which biological markers are going to be addressed and which species are going to be used.
Line 47: The term “barramundi” is exclusive to Australia. The species Lates calcarifer should be addressed by its most common name “Asian seabass”.
Lines 48-51: The journal “Molecules” is not exclusively read by Australian people, so the fish common names should be those used by most researchers in the world. The use of “red snapper” is a regional preference, and should be altered, throughout all manuscript, to “red seabream”.
Line 50: In taxonomy, the family name does not come in italics.
Line 52: To define fish specimens as “fresh” is not enough for a work that wants to assess their quality. It is important to indicate how many days have passed since the specimens were killed and began their commercialization process. It is also very important to clearly indicate if the specimens were from aquaculture, and what type of production methods were used to cultivate them (intensive vs extensive, offshore vs inshore tanks). The month in which the study was made is also important to be stated, because fatty fish species vary significantly their fat content during the year, even in aquaculture.
Lines 52-54: There is a very important issue missing in this part of the methods. When the fish were gutted, where they also stripped from their scales? The four species used in this work have completely different types (shape, sizes, and thickness) of scales. If the fish still had their scales during the SORS measurements, this fact might influence the results. If the fish did not have their scales, we must consider that the process of stripping them out can damage the skin and influence the results. None of these issues are addressed in the Methods nor in the Discussion.
Lines 55-62: More biological information about the species should be presented. For instance, what is the fat content for red seabream (red snapper)?
Lines 46-62: How many individuals were analysed from each species? This issue is of paramount importance for the publishable quality of this work. In aquaculture production, it is very common to have individuals with anatomical deformities due to inbreeding issues and large densities in tanks. These deformities are mainly structural (skeleton) but might influence the relative location of internal organs and muscle distribution, obviously influencing the molecular composition determine in certain body locations. If this study uses a very low number of individuals per species, its results might not be representative or correct for a certain species.
Line 71: Because of the recurring use of the locations where the scans were made along the manuscript, it is advisable to insert Figure S1 in the text, and not relegate it to supplementary materials.
Line 91 and line 98 (Table 1): Why did the authors focus their determinations on tryptophan, phenylalanine, guanine? Nowhere in manuscript is stated what are the relations of these compounds with seafood quality? This issue also lacks in the introduction.
Line 116: The discussion would have benefit of classical chemical analysis techniques for those compounds chosen by the authors for discussion. After making all the SORS measurements, the same specimens could have been used to perform chemical analysis methods, allowing to corroborate and validate directly all the results obtained by SORS. Instead, the authors used bibliography to discuss their results, but most of the time the references used are just for one species (mostly salmon) and should not be used for general assumption to other species.
Line 122: Reference 12 focus on salmon, how can we extrapolate to other species of different families and with variable fat content?
Lines 123-129: Data presented here is just for Atlantic Salmon. Can it be extrapolated to the other species? Probably not. To complement SORS with chemical analysis would have resolve this issue.
Lines 139-141: This potential application of SORS is completely unsupported by this work. Reference 15 correspond to a work that was done on fish oils supplements, where omega-3 fatty acids are isolated from other molecules. In whole fish, the fat is connected to the muscle proteins, and fatty acids are normally present as lipoproteins, which makes their determination and quantification much more complex and less reliable.
Author Response
Reviewer 2
General remarks:
This article is well written and has interesting results, but in my opinion has a big flaw: exploratory works that use instrumental measurements for the first time in untested species, have to validate their results using classical chemical analysis techniques. Bibliographic information determined for one species do not directly apply to others, without analytical confirmation.
“Fish” are a very diverse group of organisms, with a lot of anatomical and physiological differences. Using conclusions from studies done on salmon to infer correlations on other species can be a bit implausible, especially if we are comparing species with completely different habitats, feeding habits or anatomical features. Fish cultivated in aquaculture continue to be morphologically, anatomically, and physiologically like their wild counterparts.
In my opinion these results are publishable, but the focus, depth and scope of the article must be changed.
To facilitate the location of my comments, I am going to present them organized by line number of the submitted pdf file.
Revision comments:
- Line 2: The presented title does not describe adequately the work done in this study. When using the word “characterisation” is necessary to indicate what is going to be characterised. In this case the authors need to indicate what kind of molecules they are going to characterise. It is also necessary to indicate which species are going to be analysed in this work. The term “fish” includes a huge diversity of species, from bony fish to cartilaginous fish, and in this work only four bony species were analysed. The title should include the names of the species.
A new title has been proposed to more accurately describe the manuscript. "Detection of biomarkers relating to quality and differentiation of some commercially significant whole fish using Spatially Off-set Raman Spectroscopy"
See lines 2 – 5.
- Line 28: The Introduction is very short and lacks topics necessary to fully understand the work! The fundamentals of Raman spectroscopy need to be described, along with the explanation on how it can be applied to determine specific molecules’ contents in skin and muscle tissues. The Introduction also lacks the advancements made by other authors in this field.
Introduction now contains more background on Raman Spectroscopy along with more explanation on the application of Raman and SORS for detection of biomarkers in tissues and muscle foods.
See lines 36 – 75.
- Line 30: The citation [1] is from 2018. FAO publishes annually the State of World Fisheries and Aquaculture. Authors should use the reference from 2020.
We have changed the reference to the most recent (2020) State of World Fisheries FAO publication.
See line 275.
- Line 33: The expression “to determine” is used twice.
We have removed the duplicated expression.
- Lines 42-43: The aim of this study needs to explicitly indicate which biological markers are going to be addressed and which species are going to be used.
The wording of the aim has been changed to:
"The aim of this project was to determine the viability of SORS to detect biological markers relating to fish quality through the skin in whole fish. A particular objective was to test the ability of SORS to detect lipid and pigmentation biomarkers, as these are of significant importance in aquaculture as measures of fish quality and health. A small survey group of commercial fish was chosen in order to test the sensitivity of SORS spectra to these biomarkers, however this is still sufficient for testing the feasibility of SORS as a biomolecular characterization technique for aquaculture and fisheries.”
See lines 76 – 82.
- Line 47: The term “barramundi” is exclusive to Australia. The species Lates calcarifer should be addressed by its most common name “Asian seabass”.
Common name have been changed as recommended by the reviewer.
- Lines 48-51: The journal “Molecules” is not exclusively read by Australian people, so the fish common names should be those used by most researchers in the world. The use of “red snapper” is a regional preference, and should be altered, throughout all manuscript, to “red seabream”
Common name changed as recommended by the reviewer.
- Line 50: In taxonomy, the family name does not come in italics.
Family name is no longer italicised.
- Line 52: To define fish specimens as “fresh” is not enough for a work that wants to assess their quality. It is important to indicate how many days have passed since the specimens were killed and began their commercialization process. It is also very important to clearly indicate if the specimens were from aquaculture, and what type of production methods were used to cultivate them (intensive vs extensive, offshore vs inshore tanks). The month in which the study was made is also important to be stated, because fatty fish species vary significantly their fat content during the year, even in aquaculture.
As these samples were acquired from a local fish market, it is not possible to determine the amount the time that they spent post-mortem. The samples in this manuscript are very much "real world" samples in that they are commercial products exactly as they are available to the general public. The intended purpose of this paper was to show a proof of concept that SORS could have commercial applications based on a small survey of common market fish. Samples that are tightly controlled, or have extensively detailed history, are not necessary for this application.
Other projects are planned in the future to determine spectral differences that may arise from differences in cultivation, time spent post-mortem, etc. These projects will require more controlled or historically detailed samples as this reviewer has pointed out. However, for this present study our experimental design has demonstrated that SORS spectra can reveal biochemical information of interest to the aquaculture industry, such as our collaborative partners Petuna Aquaculture Group.
Measurements were taken in September 2019. See line 89.
- Lines 52-54: There is a very important issue missing in this part of the methods. When the fish were gutted, where they also stripped from their scales? The four species used in this work have completely different types (shape, sizes, and thickness) of scales. If the fish still had their scales during the SORS measurements, this fact might influence the results. If the fish did not have their scales, we must consider that the process of stripping them out can damage the skin and influence the results. None of these issues are addressed in the Methods nor in the Discussion.
The fish were not descaled prior to measurement, see line 88, in order to better preserve the skin and to more closely mimic the condition of live fish, an aim of future studies.
There are differences in the guanine band intensities between fish species, however it is not clear if this due to the size, shape, thickness of the scales or due to the concentration of guanine in the iridophore cells present below the skin surface. See lines 215-218.
- Lines 55-62: More biological information about the species should be presented. For instance, what is the fat content for red seabream (red snapper)?
Additional biological information for each species has been added to the manuscript, see lines 188-190, 193-196
Lines 46-62: How many individuals were analysed from each species? This issue is of paramount importance for the publishable quality of this work. In aquaculture production, it is very common to have individuals with anatomical deformities due to inbreeding issues and large densities in tanks. These deformities are mainly structural (skeleton) but might influence the relative location of internal organs and muscle distribution, obviously influencing the molecular composition determine in certain body locations. If this study uses a very low number of individuals per species, its results might not be representative or correct for a certain species.
Two of each species were analysed, this detail has been added to the Methods section. As mentioned previously, this manuscript details a proof of concept and as such large numbers of samples are not necessary to demonstrate this.
- Line 71: Because of the recurring use of the locations where the scans were made along the manuscript, it is advisable to insert Figure S1 in the text, and not relegate it to supplementary materials.
S1 has been moved into the Method section as Figure 1. Other figures have been relabelled accordingly.
- Line 91 and line 98 (Table 1): Why did the authors focus their determinations on tryptophan, phenylalanine, guanine? Nowhere in manuscript is stated what are the relations of these compounds with seafood quality? This issue also lacks in the introduction.
While these biomarkers do not necessarily relate to any indications of seafood quality, these wavebands are visible within the spectra and as such remarked upon. These are the first reports of these marker bands for these fish samples using SORS and as such are important to establish to help guide future research in this area.
- Line 116: The discussion would have benefit of classical chemical analysis techniques for those compounds chosen by the authors for discussion. After making all the SORS measurements, the same specimens could have been used to perform chemical analysis methods, allowing to corroborate and validate directly all the results obtained by SORS. Instead, the authors used bibliography to discuss their results, but most of the time the references used are just for one species (mostly salmon) and should not be used for general assumption to other species.
Lipids, carotenoids and the other biomarker samples mentioned in our manuscript have been reported in a range of different articles and for a range of different samples. Analysis to quantitate level of lipids, carotenoids, etc are planned for the next stages of our ongoing project but for this proof-of-principle study we believe that the information from the literature that we have used to support our band assignments and analyses provide the required independent validation and are consistent with standard practice.
- Line 122: Reference 12 focus on salmon, how can we extrapolate to other species of different families and with variable fat content?
The increased lipid band intensity in the belly region occurs in each of the fish species examined in this article. This has been documented in Atlantic salmon, and given our results we believe that it is not much of an extrapolation to see that belly regions in other fish species accumulate fats.
- Lines 123-129: Data presented here is just for Atlantic Salmon. Can it be extrapolated to the other species? Probably not. To complement SORS with chemical analysis would have resolve this issue.
An additional reference has been added to show that lipid storage in myosepta occurs in a number of fish species, including rainbow trout and red seabream.
http://doi.org/10.2108/zs150096
See lins 186-189.
- Lines 139-141: This potential application of SORS is completely unsupported by this work. Reference 15 correspond to a work that was done on fish oils supplements, where omega-3 fatty acids are isolated from other molecules. In whole fish, the fat is connected to the muscle proteins, and fatty acids are normally present as lipoproteins, which makes their determination and quantification much more complex and less reliable.
We agree that differentiation and quantification of lipoproteins in fish may be a complex and challenging problem. However, we emphasise that this study has demonstrated that SORS can reveal relevant biomarkers for lipids, proteins and several constituent amino acids, suggesting the potential for SORS for biochemical analysis in whole, and potentially even live, fish. While further research is obviously required to test our hypothesis, as this has not been reported before, we disagree that our hypothesis is unsupported. Other work has shown that fatty acid composition in fish tissue is possible using Raman Spectroscopy https://doi.org/10.1016/j.aca.2006.05.013 and our study indicates that SORS can provide a similar level of sensitivity to Raman bands beneath the skin of intact fish. This reference has been added to this section. We have edited this paragraph to show that we are aware of the complexity of this type of analysis. See lines 201 - 208.
Round 2
Reviewer 1 Report
Authors stated most of the points raised from previous reviewing process and now the manuscripts has been improved to the level of acceptance with some minor corrections.
- Regarding point 2, authors provided more information and background. However, authors stated that in SORS mathematically isolates the subsurface spectra. I believe SORS can obtain the subsurface spectra optically without any post processing, which be in many different formats including the one in https://doi.org/10.1002/jbio.201700129.
- A laser power of 450 mW is a quite big dosage for biological samples. What is the laser density or laser spot size? Did authors find any photodamage on the fresh fish?
Author Response
Corrections Round 2 – Minor
Reviewer 1
- Regarding point 2, authors provided more information and background. However, authors stated that in SORS mathematically isolates the subsurface spectra. I believe SORS can obtain the subsurface spectra optically without any post processing, which be in many different formats including the one in https://doi.org/10.1002/jbio.201700129.
- A laser power of 450 mW is a quite big dosage for biological samples. What is the laser density or laser spot size? Did authors find any photodamage on the fresh fish?
The laser did not cause noticeable damage to the surface of the fresh fish.
The laser power is 450mW at the source but is 5mW at the sample. This is not enough to damage the fish. This has been updated in the paper. See Line 109.
Reviewer 2 Report
No comments.
Author Response
Reviewer 2
No comments were made by this reviewer.